DATA RELEASE

# An improved chromosome-level genome assembly of perennial ryegrass (*Lolium perenne* L.)

Yutang Chen[1], Roland Kölliker[1,*], Martin Mascher[2], Dario Copetti[3], Axel Himmelbach[2], Nils Stein[2] and Bruno Studer[1,*]

1 Molecular Plant Breeding, Institute of Agricultural Sciences, ETH Zurich, Universitaetstrasse 2, 8092, Zurich, Switzerland
2 Leibniz Institute of Plant Genetics and Crop Plant Research (IPK), Corrensstrasse 3, 06466, Seeland, Germany
3 Arizona Genomics Institute, School of Plant Sciences, University of Arizona, Tucson, AZ 85721, USA

## ABSTRACT

This work is an update and extension of the previously published article "Ultralong Oxford Nanopore Reads Enable the Development of a Reference-Grade Perennial Ryegrass Genome Assembly" by Frei *et al.* The published genome assembly of the doubled haploid perennial ryegrass (*Lolium perenne* L.) genotype Kyuss (Kyuss v1.0) marked a milestone for forage grass research and breeding. However, order and orientation errors may exist in the pseudo-chromosomes of Kyuss, since barley (*Hordeum vulgare* L.), which diverged 30 million years ago from perennial ryegrass, was used as the reference to scaffold Kyuss. To correct for structural errors possibly present in the published Kyuss assembly, we *de novo* assembled the genome again and generated 50-fold coverage high-throughput chromosome conformation capture (Hi-C) data to assist pseudo-chromosome construction. The resulting new chromosome-level assembly Kyuss v2.0 showed improved quality with high contiguity (contig N50 = 120 Mb), high completeness (total BUSCO score = 99%), high base-level accuracy (QV = 50), and correct pseudo-chromosome structure (validated by Hi-C contact map). This new assembly will serve as a better reference genome for *Lolium* spp. and greatly benefit the forage and turf grass research community.

**Submitted:** 07 December 2023

\* Corresponding authors. E-mail: roland.koelliker@usys.ethz.ch; bruno.studer@usys.ethz.ch

Preprint submitted at https://doi.org/10.1101/2023.12.05.570088

**Subjects** Genetics and Genomics, Bioinformatics, Plant Genetics

## CONTEXT

Previously, a reference-grade genome assembly was obtained for Kyuss, which is a doubled haploid ($2n = 2x = 14$) perennial ryegrass (*Lolium perenne* L., NCBI:txid4522) genotype, using 30-fold coverage ultra-long Oxford Nanopore Technologies (ONT) reads with a read N50 greater than 60 kb [1]. The assembly was anchored to pseudo-chromosomes based on the gene order of barley (*Hordeum vulgare* L.), since the perennial ryegrass genome showed high synteny to the barley genome [2]. However, as these two genomes diverged from a common ancestor around 30 million years ago [2], any unnoticed structural variations between the two genomes might have resulted in order and orientation errors in Kyuss pseudo-chromosomes when the barley genome was used for scaffolding. The order and orientation errors in the published Kyuss assembly may lead to misleading results in downstream studies, and therefore, they should be corrected.

**Table 1.** Assembly quality statistics.

|  | Kyuss v1.0 | Kyuss v2.0 |
|---|---|---|
| Total size of the assembly (Gb) | 2.28 | 2.26 |
| Total size of pseudo-chromosomes (Gb) | 2.00 | 2.24 |
| Unanchored (Mb) | 277 | 23 |
| Contig N50 (Mb)/L50 (#) | 12/65 | 120/7 |
| Contig N90 (Mb)/L90 (#) | 3.30/209 | 31/20 |
| Scaffold N50 (Mb)/L50 (#) | 275/4 | 333/4 |
| Scaffold N90 (Mb)/L90 (#) | 10/11 | 267/7 |
| Number of gaps in pseudo-chromosomes | 234 | 59 |
| Base-level accuracy (QV) | 40 | 50 |
| Total complete BUSCO of contigs (%) | 96.00 | 99.00 |
| Total complete BUSCO of pseudo-chromosomes (%) | 92.40 | 98.60 |
| K-mer based completeness (%) | 99.39 | 99.48 |
| Short-read mapping rate (%) | 99.55 | 99.55 |

High-throughput chromosome conformation capture (Hi-C) has been widely used as a tool to capture the physical proximity of loci in the genome, providing long-range linkage information for scaffolding *de novo* genome assemblies [3]. Hi-C has now been routinely adopted in *de novo* genome assembly pipelines for assembly error correction and pseudo-chromosome construction [4–6], which has led to many high-quality genome assemblies for species in the grass family, such as barley [7], oat (*Avena sativa* L.) [8], rye (*Secale cereale* L.) [9] and perennial ryegrass [10].

To correct for any potential structural errors in the published Kyuss v1.0 assembly [1], in this work, we *de novo* assembled the genome with the same ONT data from Kyuss v1.0 but added about 50-fold coverage Hi-C data to assist pseudo-chromosome construction. This resulted in the improved chromosome-level assembly Kyuss v2.0.

## METHODS
### Sample information
The plant used in this study for Hi-C sequencing is a clonal replicate of the original Kyuss genotype [1], which is a doubled haploid perennial ryegrass genotype derived from *in vitro* anther culture [11].

### Hi-C library preparation and sequencing
Hi-C library preparation and sequencing were performed at IPK Gatersleben, following the procedure used for rye [9]. The restriction enzyme *DpnII* was used to digest DNA, and the library was sequenced on an Illumina NovaSeq6000 platform to produce 580 million pair-end reads (2 × 151 bp), corresponding to 50-fold sequencing coverage of the genome, assuming the genome size is 2.5 Gb [1].

### Genome assembly and polishing
NextDenovo v2.5.0 (RRID:SCR_025033) [12] was used to assemble the ONT reads from Kyuss v1.0 [1]. The resulting contigs were polished by NextPolish v1.4.1 [13] with two rounds of long-read correction using the same ONT data, followed by four rounds of short-read correction using around 60-fold coverage Illumina whole-genome sequencing (WGS) pair-end reads from Kyuss v1.0 [1]. After polishing, the assembly reached a total size of 2.26 Gb, consisting of 77 contigs with a contig N50 of 120 Mb (Table 1).



**Table 2.** Genome annotation results.

|  | Kyuss v1.0 | Kyuss v2.0 |
|---|---|---|
| Total number of genes | 38,868 | 38,765 |
| Number of genes in pseudo-chromosomes | 35,884 | 38,632 |
| Number of genes in unanchored sequences | 2,984 | 133 |

## Pseudo-chromosome construction

Scaffolding and pseudo-chromosome construction with Hi-C data were accomplished using the TRITEX pipeline [6] with a *L. multiflorum* haploid assembly as the guide map (NCBI BioProject accession number PRJNA990649) following the steps described for oat [8]. Briefly, Hi-C reads were first processed by Cutadapt v3.2 (RRID:SCR_011841) [14] to trim the chimeric 3′ end, then the cleaned reads were aligned to the assembly using Minimap2 v2.17 (RRID:SCR_018550) [15]. Alignment results were output into a binary Sequence Alignment Map format (BAM) file by SAMtools v1.11 (RRID:SCR_002105) [16], and Novosort v1.04.03 [17] was subsequently used to sort the aligned reads in the file. Pairwise Hi-C links were extracted from the sorted BAM file by BEDTools (RRID:SCR_006646) [18] and stored in a Browser Extensible Data (BED) format file. Further, Hi-C links were imported to R [19] to scaffold contigs using TRITEX scripts [20] following the instructions detailed at https://tritexassembly.bitbucket.io/. Chimeric contigs containing sequences from unlinked loci joined mistakenly by the assembler were detected and corrected based on the Hi-C contact map using TRITEX scripts. After several rounds of inspection and correction of the chimeric contig, the corrected contigs were ordered and oriented by Hi-C links to build pseudo-chromosomes. Finally, a FASTA format file with all pseudo-chromosomes and unanchored sequences was generated.

The pseudo-chromosomes constructed by TRITEX were further examined by another Hi-C scaffolding pipeline consisting of Juicer v2.0 (RRID:SCR_017226) [21], JuiceBox v1.11.08 (RRID:SCR_021172) [22], and 3D-DNA [4]. Any remaining assembly errors in the pseudo-chromosomes were manually corrected using JuiceBox. Subsequently, a new FASTA file with the improved pseudo-chromosomes and the unanchored sequences was output as the final assembly by 3D-DNA.

The final chromosome-level assembly, referred to as Kyuss v2.0, reached a total size of 2.26 Gb with a scaffold N50 of 333 Mb, and 99.0% of the assembly was anchored to seven pseudo-chromosomes, leaving only 23 Mb unanchored (Table 1).

## Genome annotation

Genome annotation was done by mapping the genes from Kyuss v1.0 to Kyuss v2.0 using Lifftoff v1.6.3 [23] with Minimap2 v2.24 [15]. In total, 99.7% of Kyuss v1.0 genes were mapped to Kyuss v2.0, and only 103 genes failed to map. Of the mapped genes, 99.6% were located in pseudo-chromosomes of Kyuss v2.0, and only 133 genes were located in unanchored sequences (Table 2). Thus, the number of genes located in pseudo-chromosomes was 7.7% higher for Kyuss v2.0 (38,632) when compared to Kyuss v1.0 (35,884; Table 2).

## Repeat annotation

Repetitive sequences in Kyuss v2.0 were annotated using RepeatMasker v4.1.2-p1 (RRID:SCR_012954) [24], following the pipeline described in Kyuss v1.0 [1] with the



**Table 3.** Proportion of transposable elements in Kyuss v2.0.

| | Number of elements | Total length (bp) | Percentage of sequence (%) |
|---|---|---|---|
| Class I: Retroelements | 530,520 | 1,183,157,557 | 52.24 |
| LINEs | 42,653 | 43,511,255 | 1.92 |
| RTE (RIT) | 2,687 | 2,444,426 | 0.11 |
| L1 (RIL) | 39,964 | 41,066,771 | 1.81 |
| SINEs | 234 | 27,095 | 0.00 |
| LTR elements | 487,633 | 1,139,619,207 | 50.32 |
| Copia (RLC) | 93,393 | 172,003,774 | 7.59 |
| Gypsy (RLG) | 380,138 | 962,763,695 | 42.51 |
| Class II: DNA transposons | 201,899 | 121,972,912 | 5.38 |
| hAT (DTA) | 16,561 | 12,610,881 | 0.56 |
| Tc1-Mariner (DTT) | 21,135 | 3,826,993 | 0.17 |
| PIF-Harbinger (DTH) | 37,483 | 17,584,919 | 0.78 |
| Helitron (DHH) | 8,587 | 1,420,581 | 0.06 |
| Unclassified | 1,298,753 | 454,417,422 | 20.06 |

customized repeat database generated for a *L. multiflorum* genotype [25]. RepeatMasker identified 52.24%, 5.38%, and 20.06% of the assembly as retroelements, DNA transposons, and unclassified repeats, respectively, suggesting that nearly 80% of the genome consists of transposable elements (Table 3).

## DATA VALIDATION AND QUALITY CONTROL

### Assembly quality assessment

The quality of Kyuss v2.0 was evaluated based on contiguity, completeness, and base-level accuracy. Quality check results were compared between the two versions to show the difference in quality between Kyuss v1.0 and v2.0.

First, the contiguity of the assembly was evaluated by basic assembly statistics, such as contig and scaffold N50/90 and L50/90 (Table 1), calculated using assembly-stats [26]. By comparing these metrics, Kyuss v2.0 was observed to be much more contiguous than Kyuss v1.0. For example, the contig N50 of Kyuss v2.0 was 120 Mb, which is ten times higher than the 12 Mb contig N50 observed in Kyuss v1.0 (Table 1), suggesting much higher contiguity of Kyuss v2.0 at the contig level. The scaffold N50 of Kyuss v2.0 was 333 Mb, which is 58 Mb higher than that of Kyuss v1.0 (Table 1), suggesting that Kyuss v2.0 is still much more contiguous at the scaffold level. Other contiguity metrics shown in Table 1 all supported the same observation.

Second, the completeness of the assembly was evaluated by different metrics, including the total length of the assembly, k-mer spectra completeness, BUSCO score, and short-read mapping rate. The total length of Kyuss v2.0 is 2.26 Gb, covering 90.4% of the genome size as estimated based on k-mers (2.5 Gb) [1], and very little difference was observed in the total length between Kyuss v1.0 and v2.0 (Table 1). However, when comparing the total length of pseudo-chromosomes between the two versions, Kyuss v2.0 is 240 Mb bigger than Kyuss v1.0 (Table 1), suggesting that pseudo-chromosomes of Kyuss v2.0 are more complete. Besides, gaps in pseudo-chromosomes were detected using SeqKit v2.3.0 [27], and many more gaps were observed in Kyuss v1.0 than v2.0 (234 compared to 59, Table 1). This observation also suggested that pseudo-chromosomes of Kyuss v2.0 are more complete.

A k-mer spectra plot (Figure 1) was generated by intersecting 23-mers between Kyuss v2.0 contigs and WGS short reads using KAT v2.4.2 [28], and a single red peak was observed, suggesting that the contigs represent a complete haploid genome. The estimated



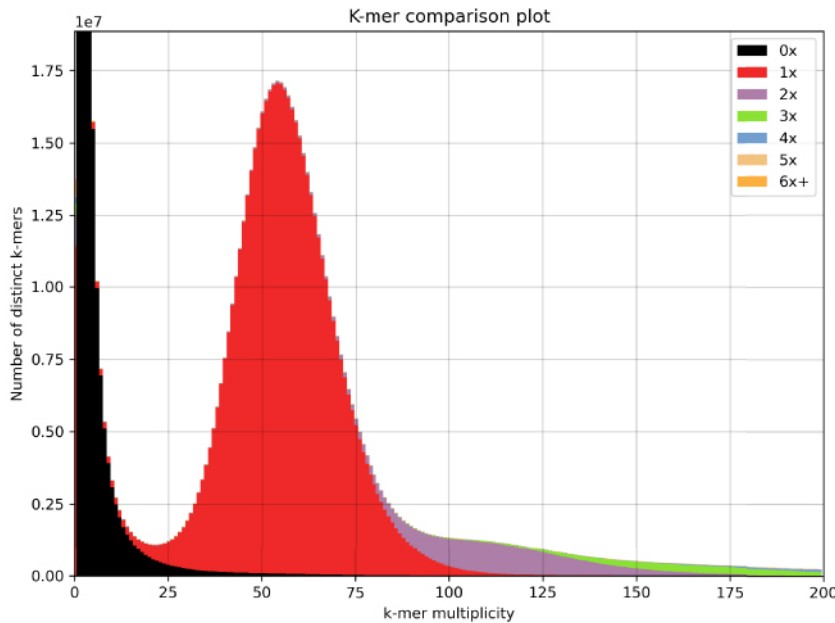

**Figure 1.** KAT k-mer comparison plot. The *x*-axis indicates the frequency of 23-mers in short reads from whole-genome sequencing, and the *y*-axis indicates the corresponding number of unique 23-mers from short reads at different frequencies. Colors, as indicated by the legend in the plot, suggest how many times a 23-mer is present in the assembly. If the genome assembly is correct, different colors should occur accordingly with increasing k-mer frequency along the *x*-axis. The black color, which means k-mer is not present in the genome assembly, should only occur at low k-mer frequency if the genome assembly is complete and represents error k-mers in the sequencing reads.

completeness given by KAT based on 23-mers was 99.5% for Kyuss v2.0 and showed little difference to Kyuss v1.0 (Table 1) [1].

BUSCO score was calculated by searching for 1,614 universal single-copy orthologs from embryophyta_odb10 database in Kyuss v2.0 contigs using BUSCO v5.3.2 (RRID:SCR_015008) [29]. About 99.0% of the orthologs were found complete in Kyuss v2.0 contigs (Table 1), with 94.9% being single-copy complete and 4.1% being duplicated complete. Only 0.6% of the orthologs were found fragmented, and only 0.4% of the orthologs were missing. The total complete BUSCO score was concordant with the estimated completeness from KAT, suggesting that Kyuss v2.0 contigs represent a very complete haploid genome. The total complete BUSCO score of Kyuss v1.0 contigs was 96.0% (Table 1), which is slightly lower than Kyuss v2.0, suggesting that Kyuss v2.0 is slightly more complete than Kyuss v1.0 at the contig level. In addition to the BUSCO analysis of contigs, the same BUSCO analysis was applied to pseudo-chromosomes of both assemblies. As a result, 88.7% single-copy complete, 3.7% duplicated complete, 0.6% fragmented, and 7.0% missing orthologs were found for Kyuss v1.0 and 94.4% single-copy complete, 4.2% duplicated complete, 0.6% fragment and 0.8% missing orthologs were found for Kyuss v2.0. Clearly, Kyuss v2.0 showed a much higher completeness when compared to Kyuss v1.0 in terms of the total complete BUSCO score of pseudo-chromosomes (98.6% compared to 92.4%, Table 1).

The short-read mapping rate was calculated using SAMtools flagstat [16], with the alignment files generated by mapping WGS short-reads to both Kyuss v1.0 and v2.0 scaffolds using BWA-MEM (RRID:SCR_022192) [30]. The same mapping rate (99.55%) was

observed between the two versions (Table 1), suggesting that the two assemblies have similar completeness if pseudo-chromosomes and unanchored sequences were counted.

To measure base-level accuracy, Polca [31] was used to check the inconsistencies between WGS short reads and both Kyuss assemblies based on the alignment of short reads to scaffolds. The quality score (QV) estimated by Polca for Kyuss v2.0 was 50, suggesting a 99.999% base-level accuracy with the probability of one sequencing error per 100 kb. The estimated accuracy of Kyuss v1.0 is 99.990% (QV 40, Table 1), which is ten times lower than Kyuss v2.0, suggesting that Kyuss v2.0 is more accurate than Kyuss v1.0.

In summary, Kyuss v2.0 is more contiguous and complete (in terms of total size anchored to pseudo-chromosomes) and has more accurate chromosome-level assembly when compared to Kyuss v1.0.

## Pseudo-chromosome structure correctness assessment

Inter-chromosome Hi-C contact maps were constructed for both versions of the Kyuss assembly to evaluate the correctness of the pseudo-chromosome structure. Hi-C data were first mapped to both assemblies using BWA-MEM [30], following the command suggested by HiCExplorer [32] (https://hicexplorer.readthedocs.io/en/latest/content/example_usage.html). Then, the Hi-C contact map of each assembly was constructed using tools from HiCExplorer such as hicFindRestSite, hicBuildMatrix, and hicCorrectMatrix, and later, the map was converted to HiC-Pro [33] format by hicConvertFormat and finally, visualized using an R script [19].

Strong Hi-C contacts were observed between pseudo-chromosomes of Kyuss v1.0, such as those between pseudo-chromosome 2 and 4, pseudo-chromosome 5 and 7 (Figure 2A), indicating that some sequences were placed to the wrong pseudo-chromosome. In addition, strong off-diagonal contacts were observed within pseudo-chromosomes for every pseudo-chromosome, suggesting that some sequences were wrongly ordered or oriented within the pseudo-chromosome. In contrast, the Hi-C contact map of Kyuss v2.0 (Figure 2B) showed a very smooth diagonal line within each pseudo-chromosome, and no indications of structural errors were observed within or between pseudo-chromosomes, suggesting that Kyuss v2.0 has the correct pseudo-chromosome structure.

## Genomic features of Kyuss v2.0

The average GC content per 1 Mb was calculated using SeqKit v2.3.0 [27], and the mean short-read alignment depth per 1 Mb was calculated using Mosdepth v0.3.3 (RRID:SCR_018929) [34] based on the alignment results from Polca during assembly quality check. In addition, the number of genes per 1 Mb was calculated based on the genome annotation results, and the proportion of repetitive sequences per 1 Mb was calculated based on the repeat annotation results. For each pseudo-chromosome, a sequence motif (AAACCCT) was searched for to detect the telomere position using quarTeT [35], and the centromeric region was estimated based on the inter-chromosome Hi-C contact using a custom R script. All these features, including pseudo-chromosome size, number of gaps, telomere and centromere positions, gene content, repeat density, GC content, and short-read alignment depth of each pseudo-chromosome were visualized in a circos plot (Figure 3A) using R package circlize (RRID:SCR_002141) [36].

With the circos plot (Figure 3A), the distribution of gaps in pseudo-chromosomes was observed (black lines, track a), and telomere sequences were identified for



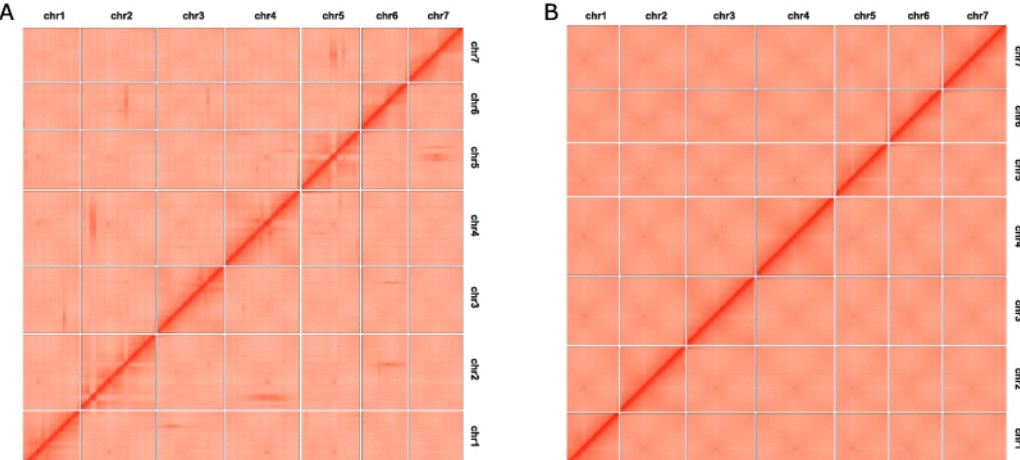

**Figure 2.** (A) Inter-chromosome Hi-C contact map of Kyuss v1.0. The contact map shows pair-wise interaction between loci within or between chromosomes. Each red pixel suggests one interaction captured between two loci. Theoretically, more interactions should be observed within chromosomes and between close loci. Therefore, for a correctly ordered and oriented assembly, most signals should be observed along the diagonal line within the pseudo-chromosome. Strong signals away from the diagonal suggest assembly or scaffolding errors. (B) Inter-chromosome Hi-C contact map of Kyuss v2.0. The smooth signal along the diagonal line within each pseudo-chromosome indicates the correct order and orientation of sequences within the pseudo-chromosome.

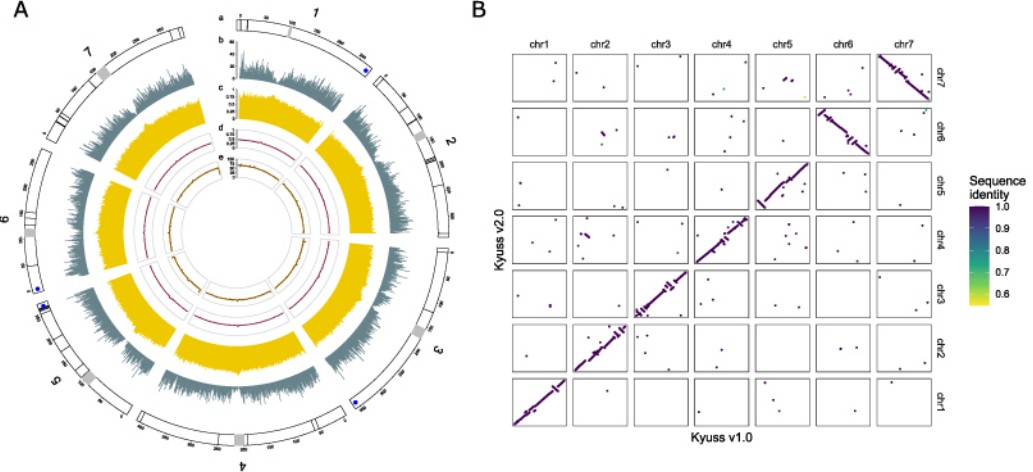

**Figure 3.** (A) Circos plot showing features of Kyuss v2.0 pseudo-chromosomes. The plot shows pseudo-chromosome length in Mb, number of gaps (black lines), telomeres (blue dots) and centromeres (gray rectangles) (a), gene counts per 1 Mb (b), the proportion of repetitive sequences per 1 Mb (c), the percentage of GC bases per 1 Mb (d) and the alignment coverage of whole genome sequencing short reads per 1 Mb (e). (B) Dot plot illustrating the synteny between Kyuss v1.0 and Kyuss v2.0. Each dot in the plot represents a gene mapped from v1.0 to v2.0, and the color of the dot specifies the gene sequence identity between two assemblies. The darker the dot, the higher the sequence identity, as indicated by the color bar on the right.

pseudo-chromosomes 1, 3, 5 and 6 at one end of the pseudo-chromosome (blue dots, track a). The rough centromeric regions estimated based on Hi-C contact were shown as gray (track a), and a significant drop in gene numbers was observed in the centromeric regions (Figure 3A, track b). The average repeat density was above 75.0% along the

pseudo-chromosome (Figure 3A, track c), and a higher repeat density (close to 1) was observed in centromeric regions (Figure 3A, track c). The average GC content along the pseudo-chromosome was 44.0% (Figure 3A, track d), and a higher GC content ranging from 46.0% to 50.0% was observed in centromeric regions. The short-read alignment depth showed a similar pattern with an expected 65-fold average coverage along the pseudo-chromosome and a higher coverage (up to 85-fold) in centromeric regions (Figure 3A, track e).

### Gene-based synteny between Kyuss v1.0 and Kyuss v2.0

A dot plot was generated to illustrate the gene-based synteny between Kyuss v1.0 and Kyuss v2.0 using LiftoffTools [37] with the annotated genes of both assemblies (Figure 3B). Many structural variations were observed between the two assemblies. For example, the orientation of pseudo-chromosome 6 and 7 was opposite in the two assemblies, and a group of genes that were in pseudo-chromosome 2 in Kyuss v1.0 was observed in pseudo-chromosome 4 in Kyuss v2.0. Since the two assemblies represent the same genome, any structural variations observed from the dot plot should not indicate real chromosome rearrangements but assembly and scaffolding errors. On top of that, as the Hi-C contact map showed that Kyuss v2.0 has the correct pseudo-chromosome structure (Figure 3B) and there were structural errors in Kyuss v1.0 pseudo-chromosomes (Figure 2A), all the structural errors indicated by the dot plot should be from Kyuss v1.0. The large difference in pseudo-chromosome structure between the two assemblies strongly suggests that an improved assembly of Kyuss (Kyuss v2.0) was needed and should be used for downstream studies to avoid misleading results.

### RE-USE POTENTIAL

In this work, we deliver an improved chromosome-level genome assembly for Kyuss, a doubled haploid perennial ryegrass genotype, using newly generated Hi-C data with ONT and Illumina sequencing data from previous work [1]. The new assembly, Kyuss v2.0, is more contiguous, more complete, and more accurate than Kyuss v1.0 [1], and most importantly, Kyuss v2.0 shows the correct pseudo-chromosome structure, whereas Kyuss v1.0 shows structural errors which could be problematic for downstream studies. We believe, with the improved quality, Kyuss v2.0 will serve as an invaluable genomic resource to yield more accurate results for downstream genomic applications, such as read mapping, variant calling, genome-wide association studies, comparative genomics, and evolutionary biology, which will greatly benefit forage and turf grass research and breeding.

### AVAILABILITY OF SOURCE CODE AND REQUIREMENTS

- Project name: Kyuss-v2.0
- Project home page: https://github.com/Yutang-ETH/Kyuss-v2.0
- Operating system(s): Platform independent
- Programming language: R/shell
- License: GNU GPL.

### DATA AVAILABILITY

The genome assembly Kyuss v2.0 and the Hi-C data are available on NCBI under BioProject accession number PRJNA1045569. All other data is available in the GigaDB repository [38].

## LIST OF ABBREVIATIONS

Hi-C: high-throughput chromosome conformation capture; ONT: Oxford Nanopore Technologies; WGS: whole-genome sequencing; BAM: binary sequence alignment/map; BED: browser extensible data; QV: quality value; kb: kilobase pairs; Mb: megabase pairs; Gb: gigabase pairs.

## DECLARATIONS

### Ethical approval

Not applicable.

### Consent for publication

The manuscript has been read and approved by all named authors.

### Competing interests

The authors declare that they have no competing interests.

### Authors' contributions

YTC, RK, BS and DC conceived this study, for which BS acquired funding. YTC conducted the bioinformatics analyses, supervised by RK, BS and MM. YTC wrote the manuscript with RK and BS. AH and NS generated the Hi-C data and, together with DC and MM, reviewed the manuscript.

### Funding

This work was supported by the European Union's Horizon 2020 research and innovation program under the Marie Skłodowska-Curie grant agreement No 847585 – RESPONSE.

### Acknowledgements

We sincerely thank Verena Knorst from the Molecular Plant Breeding group at ETH Zurich for taking care of the plant material. We sincerely thank Daniel Ariza Suarez from the Molecular Plant Breeding group at ETH Zurich for his kind help in figure preparation. We sincerely thank Ines Walde (IPK Gatersleben) for Hi-C library preparation and sequencing. We also thank ISG-HEST at ETH Zurich for providing computational resources as well as their IT service for this work.

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
