## [Editor Report]

Editor’s AssessmentThis Data Release paper presents an updated genome assembly of the doubled haploid perennial ryegrass (Lolium perenne L.) genotype Kyuss (Kyuss v2.0). To correct for structural the authors de novo assembled the genome again with ONT long-reads and generated 50-fold coverage high-throughput chromosome conformation capture (Hi-C) data to assist pseudo-chromosome construction. After being asked for some more improvements to gene and repeat annotation the authors now demonstrate the new assembly is more contiguous, more complete, and more accurate than Kyuss v1.0 and shows the correct pseudo-chromosome structure. This more accurate data have great potential for downstream genomic applications, such as read mapping, variant calling, genome-wide association studies, comparative genomics, and evolutionary biology. These future analyses being able to benefit forage and turf grass research and breeding.

---

## [Reviewer Report]

Reviewer name and names of any other individual's who aided in reviewer Qing LiuDo you understand and agree to our policy of having open and named reviews, and having your review included with the published papers. (If no, please inform the editor that you cannot review this manuscript.)YesIs the language of sufficient quality?YesPlease add additional comments on language quality to clarify if needed
NoAre all data available and do they match the descriptions in the paper? YesAdditional CommentsAre the data and metadata consistent with relevant minimum information or reporting standards? See GigaDB checklists for examples <a href="http://gigadb.org/site/guide" target="_blank">http://gigadb.org/site/guide</a>YesAdditional CommentsIs the data acquisition clear, complete and methodologically sound?YesAdditional CommentsIs there sufficient detail in the methods and data-processing steps to allow reproduction?YesAdditional CommentsIs there sufficient data validation and statistical analyses of data quality? YesAdditional CommentsIs the validation suitable for this type of data?YesAdditional CommentsIs there sufficient information for others to reuse this dataset or integrate it with other data?YesAdditional CommentsAny Additional Overall Comments to the AuthorThis updated double haploid perennial ryegrass (Lolium perenne L.) showed contig N50 of 120 Mb, total BUSCO score=99%, which verified that the improved assembly can serve a reference for Lolium species using 50-fold coverage Hi-C data. The article is well edited except for below revision points. The minor revision is suggested for the current version. 1 Please elucidate the Kyuss v2.0, whether its reference is the same as Kyuss v1.0, if same or separate reference please elucidate. 2 In Table 3 of page 6, What the repeat element number for each family, could authors listed in number and proportion in order to clear the family category, for example, is the number of rolling-circles the same for Heltrons? 3 Tandem repeat or satellite or centromere location data, could author provide for the updated assembly of the Lolium species. 4 For Figure 1, what the heterozygosity and k-mer estimated genome size, I can’t find the data. 5 In Figure 3A, lowercase letter a, b, c , d and e are suggested to subsittute the A, B, C, D and E in order to avoid Figure 3A and Figure 3AA
RecommendationMinor Revision

---

## [Reviewer Report]

Reviewer name and names of any other individual's who aided in reviewer Istvan NagyDo you understand and agree to our policy of having open and named reviews, and having your review included with the published papers. (If no, please inform the editor that you cannot review this manuscript.)YesIs the language of sufficient quality?YesPlease add additional comments on language quality to clarify if needed
Are all data available and do they match the descriptions in the paper? NoAdditional CommentsMinor revision in the manuscript body is suggested. Gene annotation and repeat annotation data need some minor revision) See details in the "Additional Comments" section. Are the data and metadata consistent with relevant minimum information or reporting standards? See GigaDB checklists for examples <a href="http://gigadb.org/site/guide" target="_blank">http://gigadb.org/site/guide</a>YesAdditional CommentsIs the data acquisition clear, complete and methodologically sound?YesAdditional CommentsIs there sufficient detail in the methods and data-processing steps to allow reproduction?YesAdditional CommentsIs there sufficient data validation and statistical analyses of data quality? YesAdditional CommentsIs the validation suitable for this type of data?YesAdditional CommentsIs there sufficient information for others to reuse this dataset or integrate it with other data?YesAdditional CommentsThe submitted dataset reports and improved chromosome-level assembly and annotation of the doubled-haploid line Kyuss of Lolium perenne. The present v2.0 assembly is showing significant improvements as compared to the Kyuss v1.0 assembly published by the same group in 2021: The new assembly incorporates 99% of the estimated genome size in seven pseudo-chromosomes and the >99% BUSCO completeness of the gene space is also impressive.  Below are mine remarks and suggestions to the present version of manuscript:  Genome assembly and polishing It's indicated that for the primary assembly of the present work the same source of ONT reads were used as for the previous Kyuss v1.0 assembly. However, in the present manuscript the authors report clearly better assembly quality as opposed to the Kyuss v1.0 assembly. The question remains open, whether the authors achieved better results by changing/optimizing the primary assembly parameters, and/or applying a step-wise, iterating strategy with repeated rounds of long-read and short-read corrections? By any means, a more detailed description/specification of assembly parameters would be desirable.   Genome annotation In the provided annotation file "kyuss_v2.gff" in the majority of cases gene IDs consisting of the reference chromosome ID and of an ongoing number, like 
"KYUSg_chr1.188" are used. However in a few cases gene IDs like "KYUSt_contig_1275.207" are also used. This inconsistency might create confusions for future users of Kyuss_2 resources, and while the later type of gene IDs might be useful for internal usage, they became meaningless, as instead of contigs now pseudo-chromosomes (and some unplaced scaffolds) are used as references. The authors should modified the gff files and use a consistent naming scheme for all genes. Further, transcript DNA sequences as well as transcript protein sequences with consistent naming schemes should also be provided.   Repeat annotation The authors should modify Table 3 by specifying and breaking down repeat categories according to the Unified Classification System of transposable elements, by giving Order and Superfamily specifications (like LTR/Gipsy and LTR/Copia etc, in accord with the provided gff file "kyuss_v2_repeatmask.gff").   According to the provided repeat annotation BED file, more than 750K repeat features have been annotated on the Kyuss_2 genome. Of these repeat features 57815 are overlapping with gene features and 25843 of these overlaps are longer than 100 bp. This indicate that a substantial portion of the 38765 annotated genes might represent sequences coding for transposon proteins and/or transposon related ORFs. I suggest that the authors revise the gene annotation data (and at least remove gene annotation entries that show ~100% overlap with repeat features).  Assembly quality assessment
"The quality score(QV) estimated by Polca for Kyuss v2.0 was 50, suggesting a 99.999% base-level accuracy with the probability of one sequencing error per 100 kb. The estimated accuracy of Kyuss v1.0 is 99.990% (QV40, Table 1), which is 10 times lower than Kyuss v2.0, suggesting that Kyuss v2 is more accurate than Kyuss v1.0." In my opinion, this sentence needs clarification as readers might have difficulties to properly interpret this - especially considering the facts that the same long-read data was used for both for the v1 as well a for the v2 assembly versions, the short-read mapping rate was the same (99.55%) for both versions and the K-mer completeness analysis results differed only slightly (99,39% vs. 99.48%).
Any Additional Overall Comments to the AuthorThe submitted dataset reports and improved chromosome-level assembly and annotation of the doubled-haploid line Kyuss of Lolium perenne. The present v2.0 assembly is showing significant improvements as compared to the Kyuss v1.0 assembly published by the same group in 2021: The new assembly incorporates 99% of the estimated genome size in seven pseudo-chromosomes and the >99% BUSCO completeness of the gene space is also impressive.  Below are mine remarks and suggestions to the present version of manuscript:  Genome assembly and polishing It's indicated that for the primary assembly of the present work the same source of ONT reads were used as for the previous Kyuss v1.0 assembly. However, in the present manuscript the authors report clearly better assembly quality as opposed to the Kyuss v1.0 assembly. The question remains open, whether the authors achieved better results by changing/optimizing the primary assembly parameters, and/or applying a step-wise, iterating strategy with repeated rounds of long-read and short-read corrections? By any means, a more detailed description/specification of assembly parameters would be desirable.   Genome annotation In the provided annotation file "kyuss_v2.gff" in the majority of cases gene IDs consisting of the reference chromosome ID and of an ongoing number, like 
"KYUSg_chr1.188" are used. However in a few cases gene IDs like "KYUSt_contig_1275.207" are also used. This inconsistency might create confusions for future users of Kyuss_2 resources, and while the later type of gene IDs might be useful for internal usage, they became meaningless, as instead of contigs now pseudo-chromosomes (and some unplaced scaffolds) are used as references. The authors should modified the gff files and use a consistent naming scheme for all genes. Further, transcript DNA sequences as well as transcript protein sequences with consistent naming schemes should also be provided.   Repeat annotation The authors should modify Table 3 by specifying and breaking down repeat categories according to the Unified Classification System of transposable elements, by giving Order and Superfamily specifications (like LTR/Gipsy and LTR/Copia etc, in accord with the provided gff file "kyuss_v2_repeatmask.gff").   According to the provided repeat annotation BED file, more than 750K repeat features have been annotated on the Kyuss_2 genome. Of these repeat features 57815 are overlapping with gene features and 25843 of these overlaps are longer than 100 bp. This indicate that a substantial portion of the 38765 annotated genes might represent sequences coding for transposon proteins and/or transposon related ORFs. I suggest that the authors revise the gene annotation data (and at least remove gene annotation entries that show ~100% overlap with repeat features).  Assembly quality assessment
"The quality score(QV) estimated by Polca for Kyuss v2.0 was 50, suggesting a 99.999% base-level accuracy with the probability of one sequencing error per 100 kb. The estimated accuracy of Kyuss v1.0 is 99.990% (QV40, Table 1), which is 10 times lower than Kyuss v2.0, suggesting that Kyuss v2 is more accurate than Kyuss v1.0." In my opinion, this sentence needs clarification as readers might have difficulties to properly interpret this - especially considering the facts that the same long-read data was used for both for the v1 as well a for the v2 assembly versions, the short-read mapping rate was the same (99.55%) for both versions and the K-mer completeness analysis results differed only slightly (99,39% vs. 99.48%).
RecommendationMinor Revision